# Transforming Healthcare Analytics with FHIR: A Framework for Standardizing and Analyzing Clinical Data

**DOI:** 10.3390/healthcare11121729

**Published:** 2023-06-13

**Authors:** Muhammad Ayaz, Muhammad Fermi Pasha, Tahani Jaser Alahmadi, Nik Nailah Binti Abdullah, Hend Khalid Alkahtani

**Affiliations:** 1Malaysia School of Information Technology, Monash University, Bandar Sunway 47500, Selangor, Malaysia; muhammad.fermipasha@monash.edu (M.F.P.); nik.nailah@monash.edu (N.N.B.A.); 2Department of Information Systems, College of Computer and Information Sciences, Princess Nourah bint Abdulrahman University, P.O. Box 84428, Riyadh 11671, Saudi Arabia; hkalqahtani@pnu.edu.sa

**Keywords:** data analytics, data analysis, FHIR, EMR, EHR

## Abstract

In this study, we discussed our contribution to building a data analytic framework that supports clinical statistics and analysis by leveraging a scalable standards-based data model named Fast Healthcare Interoperability Resource (FHIR). We developed an intelligent algorithm that is used to facilitate the clinical data analytics process on FHIR-based data. We designed several workflows for patient clinical data used in two hospital information systems, namely patient registration and laboratory information systems. These workflows exploit various FHIR Application programming interface (APIs) to facilitate patient-centered and cohort-based interactive analyses. We developed an FHIR database implementation that utilizes FHIR APIs and a range of operations to facilitate descriptive data analytics (DDA) and patient cohort selection. A prototype user interface for DDA was developed with support for visualizing healthcare data analysis results in various forms. Healthcare professionals and researchers would use the developed framework to perform analytics on clinical data used in healthcare settings. Our experimental results demonstrate the proposed framework’s ability to generate various analytics from clinical data represented in the FHIR resources.

## 1. Background

To provide a comprehensive idea to the readers about the applications of data analytics in the healthcare industry. In this section, we introduced the data analytics concept employed in the healthcare sector. Furthermore, we thoroughly discussed the data analytics concept in the clinical data represented in the healthcare latest data standard Fast Healthcare Interoperability Resource (FHIR).

### 1.1. Healthcare Data Analytics

Healthcare data analytics is the process of analyzing and interpreting large sets of healthcare data to gain insights and improve healthcare outcomes. It involves using a range of analytical techniques and tools to process data from various sources, such as electronic health records (EHRs), electronic medical records (EMRs), medical devices, claims data, patient-generated data, etc.

The rapid advancements in hardware and software technologies in recent years have ushered in a new era of data collection and processing, resulting in remarkable progress in the field of healthcare data analytics. In the realm of healthcare organizations, clinical data serve a dual purpose. Firstly, it is utilized for the delivery of healthcare services to patients. Secondly, it is used for secondary purposes such as research, analysis, quality improvement, and more. In particular, the secondary use of clinical data has emerged as a critical component of healthcare data analytics. This has resulted in a paradigm shift in recent healthcare settings, where the secondary use of healthcare data is deemed just as important as its primary use.

Electronic health record systems are leveraged to facilitate the secondary use of healthcare data, for activities such as quality improvement, safety measurement, payments, provider certification, marketing, and research [1]. Moreover, the secondary use of healthcare data has the potential to significantly enhance the healthcare experiences of individuals. It can facilitate the learning of appropriate diseases and effective treatments, deepen people’s knowledge and understanding of the effectiveness and efficiency of healthcare systems, and aid in supporting public health initiatives [1]. However, the secondary use of healthcare data also raises complex ethical, social, and technical issues; for example, questions regarding data ownership and access privileges continue to challenge the field [2].

The healthcare industry has witnessed a remarkable surge in the volume of healthcare data in recent times, primarily driven by the widespread adoption of electronic health record (EHR) systems worldwide [3]. In addition, there has been an unprecedented growth in other types of healthcare data, such as genome sequencing and other biological structures [4]. The analysis of this clinical data is commonly referred to as analytics or healthcare data analytics, which falls under the category of secondary use of clinical data. While the term data analytics is extensively used in and outside of healthcare [3], our focus in this study is on its application in the healthcare industry.

Analytics has been deployed across various domains, including healthcare. However, experts from different fields offer diverse definitions of analytics. Nonetheless, the ultimate objective of analytics, as perceived by all experts, remains consistent. Data analytics experts characterize analytics as “the comprehensive exploitation of data, statistical and quantitative analysis, explanatory and predictive models, fact-based management to drive decisions, actions, and much more” [5]. Similarly, IBM defines analytics as “the methodical use of data and associated business insights developed through applied analytical disciplines (e.g., statistical, predictive, contextual, quantitative, cognitive, and other models) to drive evidence-based decision making for planning, management, measurement, and learning. Analytics can be descriptive, predictive, or prescriptive” [6].

Moreover, the two eminent healthcare data analytics experts, Adams and Klein, outline three distinct levels and applications of analytics in the healthcare domain [7]. Each level is associated with increasing functionality and value

**Descriptive:** Refers to standard reporting types that depict current situations and problems.**Predictive:** Refers to simulation and modeling techniques that forecast trends and anticipate the outcomes of implemented actions.**Prescriptive:** Concerns financial, clinical optimization, and other outcomes.

All three levels of healthcare data analytics are of paramount importance. However, predictive analytics has gained more attention in the current healthcare landscape [3], as medical experts seek to predict various clinical-related variables in healthcare data to enhance healthcare delivery services and optimize health and financial outcomes.

With the advent of digital medical records, hospitals and other healthcare organizations are accumulating vast amounts of data at an unprecedented rate. The clinical data captured by these organizations take multifarious forms, ranging from structured data (such as laboratory results and images) to unstructured data (such as textual notes comprising clinical narratives, reports, and various other documents). For example, the well-known US healthcare company Kaiser-Permanente has a current data store for over nine million members that surpasses a staggering 30 petabytes of data [8]. Another notable example is the American Society for Clinical Oncology (ASCO), which is developing its Cancer Learning Intelligence Network for Quality (CancerLinQ) [9]. The clinical data accumulated by CancerLinQ serve myriad healthcare data analytics purposes, providing clinicians and researchers with an extensive platform for EHR data collection, data mining, and visualization, as well as the application of clinical decision support, among others.

The ultimate goal of healthcare data analytics is to use data to make informed decisions and identify patterns and trends that can help improve patient outcomes, optimize operational efficiency, and reduce costs. By analyzing data, healthcare providers can identify areas for improvement, predict health outcomes, and personalize care for individual patients.

Some common applications of healthcare data analytics include population health management, clinical decision support, disease surveillance and monitoring, and quality improvement initiatives. The field of healthcare data analytics is constantly evolving as new technologies and approaches emerge, and it is a critical area of focus for healthcare organizations looking to improve their performance and deliver better care to patients.

To summarize, data analytics has become a pivotal aspect of current healthcare settings, a core requirement for both the industry and its experts [4]. Moreover, the future of healthcare holds tremendous promise when it comes to data analytics. With the burgeoning volume of clinical and research data, coupled with the methods employed to analyze and put it to use, there is tremendous potential for improving healthcare delivery, personal health, and biomedical research. However, there is also a continuing need to improve the quality of clinical data and conduct research aimed at demonstrating how best to apply data analytics to address healthcare challenges.

### 1.2. Healthcare Data Analytics Using FHIR Data Standard

FHIR is the latest healthcare data standard that is gaining popularity in the healthcare sector [10]. FHIR provides a standardized way to represent and exchange healthcare information electronically [11]. This avant-garde standard has captured the imagination of healthcare providers due to its unparalleled ability to reduce the costs of interoperability and its potential to catalyze a new ecosystem of third-party applications [12]. FHIR’s revolutionary interoperability capabilities have surpassed the antiquated data standards of yore, such as HL7 (v2, v3, CDA).

In a recent survey conducted by Australian and New Zealand healthcare executives, the adoption of FHIR was found to increase interoperability from a measly 11% to a staggering 66% [13]. Consequently, its adaptable nature for data exchange is increasing at a rapid pace within the healthcare industry as it garners favor among stakeholders for data exchange. The survey further revealed that 55% of healthcare providers are willing to make the shift to a FHIR-based interoperability platform. Additionally, it is estimated that FHIR will be widespread in the world healthcare industry by 2024 [14]. This showed the popularity of FHIR-based interoperability in the healthcare industry and healthcare providers’ interest in its adaptability.

However, the healthcare industry’s needs go beyond mere clinical data exchange. Clinical data need to be processed for other purposes, such as data analysis, data analytics, research, and so forth. Thus, the clinical data represented in the FHIR standard need to fulfill these requirements. FHIR’s adoption is expected to increase data availability for analytics and solve the data exchange and analytics problems faced by the healthcare industry [13]. Nevertheless, the adoption of FHIR in the analytics domain remains relatively low, as the standard is still young [15]. Moreover, the tools supporting FHIR data analytics are still relatively immature [16]. However, the healthcare providers argue that they are not only interested in sharing clinical data across healthcare organizations to improve data interoperability but are more excited to process clinical data for other purposes, such as data analysis and research, to provide real-time medical services to patients. Therefore, the tools provided these services are essential in the healthcare industry.

On the flip side, the cutting-edge FHIR standard for patient clinical information presents plenty of new opportunities for visualizing, analyzing, and automating various types of healthcare data. With each passing day, fresh use cases for FHIR data analytics are building in the healthcare industry, such as real-time alerts for patient satisfaction, identifying patterns in patients’ medical records across datasets, real-time visibility into patient readmission rates, cost savings while upholding top-notch care quality, and countless more [17,18,19,20]. However, analyzing and implementing these use cases can prove challenging owing to the young stage and practicality of FHIR.

To facilitate data processing and exchange, FHIR employs REST APIs. Nonetheless, for the domain of FHIR data analytics, the FHIR APIs must possess a dynamic nature regarding data queries and processing. As data analytics are based on diverse types of data housed in varied FHIR resources, the FHIR APIs must query this data in various ways to enable effective data analysis. Additionally, FHIR has accelerated the swift delivery of a massive volume of new healthcare applications that can integrate with electronic health records (EHRs) or electronic medical records (EMRs) data via the FHIR APIs. However, most of these applications are limited to perusing data relevant to a single patient [15]. One contributing factor, among many others, could be that the FHIR APIs are not optimally suited to queries that aggregate and categorize data across a vast clinical dataset.

A related and parallel trend within the realm of health information systems involves investing in higher-quality structured data via the coding of clinical records at the point of care. With the implementation of electronic medical records (EMRs), healthcare providers are now able to incorporate a multitude of concepts into medical records using advanced terminologies such as ICD 10, LOINC, and SNOMED CT [21,22]. This affords the opportunity for more detailed analysis by enabling access to specific clinical concepts as well as the ability to query the ontology based on additional attributes and relationships to other clinical concepts.

While this technique is highly effective when analyzing clinical data based on specific codes or terminologies, it proves to be less fruitful in general concept analysis. Therefore, other scenarios, including modifications to FHIR APIs, must be considered to enable various ways of analyzing medical data for deep clinical data analysis. However, this technique is extremely challenging and requires an individual with extensive skill and experience to change the core implementation mechanisms of FHIR APIs.

Currently, the level of expertise required to make the best use of FHIR and other clinical terminology within a data analysis workflow is relatively rare in the healthcare domain [23,24,25]. Nonetheless, the applications of data analytics and analysis in healthcare settings using the FHIR data standard are a new concept and have scarcely been applied. However, due to the rapid adoption of FHIR for medical data exchange, data analytics and analysis are now a core demand of the healthcare industry to process patient medical data in various ways and provide real-time medication to improve healthcare delivery. In summary, the standardization of healthcare data plays a crucial role in clinical and translational data analysis systems, especially when large-scale data are involved. Moreover, healthcare applications for clinical statistics and analysis can significantly enhance healthcare by connecting clinical data with analytic tools, thereby engaging practitioners or clinicians in the process of medical data analysis [26,27].

Thus, in response to the pressing need to address the complex and multifaceted challenges of data analytics in the healthcare industry, this research study puts forth a cutting-edge and innovative FHIR standard-based data analytics framework. This platform is designed to tackle the healthcare industry’s data analytics issues and provide them with a scalable, standards-based data model. At present, this pioneering framework is tailored to work with workflows specifically designed for patient clinical data originating from two distinct hospital information systems: patient registration systems and laboratory information systems. Other possible data analysis workflows and customized research scenarios on the patient data from other hospital information systems could be performed on FHIR-based data but are not currently directly supported by our framework without any modification.

The developed framework utilizes a FHIR database as its dataset, with FHIR RESTful APIs that query different types of FHIR resources from the database algorithmically. The mapping algorithm and analytic engine then process the retrieved data and generate various data analytics from patient clinical data, presenting the results to end-users via a user-friendly interface.

In short, this research study provides a state-of-the-art solution for healthcare data analytics, offering healthcare professionals an innovative platform to conduct data analysis on clinical data using FHIR. With the FHIR Data Analytics Framework, healthcare professionals can now extract meaningful insights from patient data and leverage these insights to enhance patient care delivery, promote better health outcomes, and drive healthcare industry advancements forward.

This research work has three main contributions: First, the entire framework and workflow design follow the FHIR data standard, which could be reused for any other clinical data domains and could provide support for any clinical data that follow the FHIR standard. Second, the data analysis workflow and tools incorporate the experience of clinical researchers and statisticians, which could provide a starting point for FHIR researchers in this cutting-edge standard. Third, the intelligent mapping algorithm is artfully designed to facilitate the sublime process of data analytics or data analysis within the realm of FHIR-based data. The mapping algorithm could be reused for any other clinical data that follow the FHIR specification and need to process the FHIR-based data for other purposes, such as research, developing an artificial intelligence (AI) model or machine learning (ML) model, etc.

The FHIR Data Analytics Framework comprises six layers: the FHIR database, the FHIR query engine layer, the mapping algorithm/agent layer, the FHIR-compliant database layer, the analytics engine layer, and the user interface. The rest of this manuscript is structured accordingly. Section 2 provides a comprehensive literature review, while Section 3 discusses the five major materials used in this study. In Section 4, the framework’s architecture is described in detail, followed by the implementation details in Section 5. Section 6 explains the experiment setup, and Section 7 discusses the results. Section 8 describes the limitations, and Section 9 explains the discussion and future plans. Finally, Section 10 analyzes the conclusion.

## 2. Literature Review

Throughout the years, financial and administrative data were deemed essential attributes for planning purposes. However, in recent times, comprehensive healthcare data have become crucial to institutional strategic planning and self-analysis [28]. The healthcare industry heavily relies on various data sources, such as Electronic Health Record Analysis (EHRA), Biomedical Image Analysis (BIA), Sensor Data Analysis (SDA), Biomedical Signal Analysis (BSA), Genomic Data Analysis (GDA), Clinical Text Mining (CTM), and analytics methods to process and analyze clinical data [29]. Analyzing and performing data analytics on clinical data in healthcare settings is a fundamental requirement in the healthcare industry. Despite this, the literature scarcely acknowledges the use of data analytics in the healthcare industry.

In our thorough literature review, we noticed some efforts that utilized various clinical data sources in the data analytics domain. For example, the Observational Health Data Sciences and Informatics (OHDSI) program has generated an enormous volume of work in the field of health data analytics, including the creation of the Observational Medical Outcomes Partnership (OMOP) Common Data Model (CDM) [30]. The OMOP provides a target data model for health data analytics, along with analytic routines and common vocabularies that could be run over the common data model.

Furthermore, the OMOP has a rich ecosystem of applications that have been developed to assist in its implementation and use, such as the user interface designed by the authors in the [31] study to facilitate analytic queries over the OMOP data model. Moreover, researchers have also explored the use of the OpenEHR model within health data analytics, as exemplified by the work of Chunlan et al. [32] in developing the Archetype Query Language (AQL), which is a standard way of querying data from OpenEHR-based systems [33]. The AQL has been implemented in many electronic health records (EHRs) and analytics software tools and provides important design features for this type of capability [15].

However, while these attempts have been applied to EHR datasets, the application of such techniques to data represented in the Fast Healthcare Interoperability Resources (FHIR) standard is a relatively new and challenging concept. Therefore, the researchers are looking for new techniques with which they can apply data analytics to the clinical data represented in the FHIR-based standard. However, as aforementioned, the FHIR is a young data standard [15], and limited research related to FHIR analytics has been reported [34]. A recent scientific literature review study reveals that only a few studies have been reported in the literature that discussed FHIR analytics [16]. Thus, the concept of FHIR data analytics is extremely new, and so far, the state of FHIR analytics is at an early stage. Therefore, applying data analytics or data analysis is challenging and an extremely new concept in this domain. However, some researchers have made some initial efforts in FHIR-based analytical circumstances, such as the prediction of sepsis based on the FHIR standard in the [35] study and the deployment of clinical predictive models via FHIR in Web Services explained in the [36] study.

Furthermore, the use of FHIR to store and analyze medical data on a large scale has also been implemented by the Google cloud platform (GCP) and Azure, which integrated FHIR into their cloud platforms [21,26]. In addition, the tech company Startups has recognized the analytical capabilities of FHIR and utilized the doc.ai application to provide personalized medicine, automate the process of controlling audit files, and store data in a structured way [37].

Moreover, FHIR was used to support clinical decisions and to build a distributed phenotyping analytics platform [38]. Kreuzthaler et al. discussed the use and benefits of standardized data in analytical approaches [39]. In addition, Franz et al. developed a monitoring system with the FHIR data standard [40]. Liu et al. [41] explained many ways to make bulk FHIR data available for analytic queries. The authors concluded that Apache Parquet [42] is the ideal tool for storing and querying FHIR data in the context of large-scale analytics using Apache Spark.

Grimes et al. [15] discussed the use of FHIR data analytics using the pathling concept. However, it works in a limited domain because some operations are not easily or even currently possible to achieve via the FHIR REST API specification, such as data aggregation, searching the data, etc. Therefore, it is extremely challenging to implement. Furthermore, Dunn et al. [43] explained genomic data analysis using FHIR in a cloud framework. However, it only applies to the analysis of genomic data using a cloud framework and would be challenging to apply to clinical data represented in FHIR and implement in traditional healthcare settings. Similarly, Gruendner et al. [44] described the FHIR data formatting for statistical analysis. However, this technique only generated the FHIR data but failed to provide any platform for clinical data analysis or data analytics using REST APIs. Therefore, it is extremely challenging to generalize the concept and provide a platform for medical software developers and researchers to perform any data analytics on the clinical data or use the resulting data for research purposes.

Moreover, the famous health information technology services provider Cerner Corporation produces the Bunsen [45] library that encodes FHIR resources within Apache Spark [46] datasets. This work facilitates loading, transforming, and analyzing FHIR data. Cerner Corporation has also been involved with the Structured Query Language (SQL) on the FHIR proposal [47], which is a projection of the FHIR data model onto the relational query model and SQL language. Additionally, Google also discussed and implemented a method for encoding FHIR data using Buffers Protocol [48]. Furthermore, Google also developed many tools and techniques [49] for using FHIR with the BigQuery analytics platform, integrating with the FHIR Bulk Data API, and using FHIR data within cloud-based data processing and machine learning pipelines.

Despite the various initial attempts at data analytics on clinical data represented in the FHIR data standard, there has been no user-friendly data analytics framework or visualized tool to help healthcare users such as practitioners, providers, and patients perform various data analytics on patient clinical data. To address this gap, our research study developed a framework with a user-friendly interface that enables healthcare practitioners, providers, and patients to perform data analytics on the clinical data used in two hospital information systems and represented in the FHIR-based standard.

## 3. Materials

In this section, we are discussing various materials that will help us develop our framework. This information is helpful for the readers to know about the challenges and framework pre-development procedures involved in this undertaking.

### 3.1. Required Outcomes

Our ambition was to develop a data analytics framework that could perform various types of analytics on clinical data typically used in healthcare facilities and represented in the FHIR standard. Our esteemed endeavor has borne fruit, and we have proudly fashioned a data analytics framework for healthcare environments, performing an array of analytical procedures on clinical data and elegantly visualizing the resulting insights.

### 3.2. User Research and Inputs

As previously discussed, the Fast Healthcare Interoperability Resources (FHIR) is a young data standard, and the support for FHIR data analytics is still in its infancy. Moreover, the clinical data flow in healthcare settings and the data analytics concept on clinical data represented in the FHIR format remain unclear at this stage. Particularly for individuals outside of the medical field, comprehending this concept can prove to be a challenging task. Therefore, to better grasp the FHIR data analytics concept and its workflow in the healthcare environment, we decided to take input from various professionals working in healthcare settings. We conducted numerous interviews with doctors, practitioners, patients, pharmacists, and others in the healthcare industry to obtain a more comprehensive understanding of workflow and user requirements. These interviews consisted of both open-ended and closed questions related to the current challenges within the healthcare data analytics domain. Furthermore, we sought to understand the data analytics needs of various stakeholders, including patients, practitioners, and healthcare providers, regarding the healthcare industry.

This process helped us validate our assumptions about adopting FHIR data analytics in the healthcare industry and provided insight into the views of users (practitioners, patients, providers, etc.) regarding the adoption of FHIR data analytics, as well as their opinions on workflow with this new technique in this domain. It also identified a range of use-case scenarios for various analytics that we could implement in this prototype. Based on what we learned from this process, we selected the following two parameters (use cases) to serve as the focal point of our work:

**Patient cohort selection:** The selection and retrieval of patient information/records based on complex inclusion and exclusion criteria.

**Data preparation:** Processing and reshaping data in preparation for use with statistical models or tools.

### 3.3. Challenges

The FHIR is a highly nested, complex, and graph-like data format that represents clinical data in a resource structure in JSON/XML format. With a hierarchical tree structure, the data elements are nested, making it difficult to represent within traditional relational data models, especially when simplifying query logic is a primary goal. However, representing the data in a traditional relational data model is essential for data analytics and analysis. However, the graphical FHIR resource structure poses a significant challenge, and optimizing the data structure for analytics and analysis queries across a wide range of use cases also raises performance issues.

To handle these challenges, we have developed a cutting-edge mapping algorithm/agent to convert FHIR resource data into a sample EMR format and store it in a relational data model before conducting any data analytics. Our mapping algorithm was used to transform the clinical data stored in FHIR resources into a relational data model.

### 3.4. The Clinical Data Analysis Workflow Design

Performing data analytics on the data present in the dataset is challenging, as it relies totally on workflows (business use-case scenarios). The design of such workflows is quite difficult, particularly for non-medical experts, because they have issues identifying various parameters for the clinical data used in the healthcare settings, which include user requirements, data constraints, and more. Therefore, we had discussions with medical experts and, on the basis of their inputs and our common clinical data analysis requirements, we designed two general analysis workflows: patient-centered data analysis and cohort-based data analysis. Furthermore, we elaborated on the workflows and designed five primary workflows that are used in healthcare settings on the patient data and are suitable for performing data analytics on our dataset (see Table 1).

The patient-centered data analysis workflow facilitates the browsing of various pieces of information focused on the individual. Patient-specific data derived from multiple sources are integrated into a single identifier. In the FHIR data model, the patient is an independent resource, while other resources such as observation and practitioner have a property “subject” that links them to a specific patient object, representing patient-centered relationships.

The cohort-based analysis workflow refers to more common data analysis needs in clinical statistics and studies. In this workflow, the Condition/AllergyIntolerance/Observation/Practitioner of a cohort is largely measured by the distribution of patient characteristics in different dimensions. The workflow is designed to support a wide range of clinical data analysis tasks, including patient registration analysis, patient allergy timeline analysis, patient laboratory test analysis, cohort gender/age distribution statistics, and more. Overall, our workflows provide a robust framework for performing data analytics on healthcare datasets.

### 3.5. FHIR REST APIs Working Mechanism

The Fast Healthcare Interoperability Resource (FHIR) specification defines standard REST APIs to exchange a variety of healthcare data and perform a range of operations on the clinical data represented in the FHIR resource structure. These APIs are also known as the core FHIR REST APIs. The power of these APIs lies in their use of the widely accepted HTTP (GET, POST, PUT, DELETE) protocol to perform pre-defined operations such as CRUD (Create, Read, Update, Delete) on any FHIR resource. For example, with just a few clicks, one can access and retrieve the update history, view information, delete, create, or update any instance of a FHIR resource. Figure 1 illustrates the view of these operations.

Furthermore, every FHIR API conforms to a common signature and format, ensuring that FHIR-compliant systems can retrieve specific healthcare data using the same API signature and format. For example, to retrieve patient demographic information based on the patient’s name and date of birth, one can use the following API:

GET http://baseURL/Patient?given=[patient given name]&birthDate=[date of birth]

This API will retrieve the patient’s name and date of birth. In this API, “Patient” is the FHIR patient resource, while “given” and “birthDate” are the given parameters. The output of this API will be in standard JSON/XML format, with tags and elements following strict standards. Figure 2 provides an illustrative view of a sample API operation mechanism in which the APIs access the clinical data, and we performed data analytics on that data.

## 4. FHIR Data Analytics Framework

We developed an FHIR data analytics framework used to perform various data analytics on the clinical data represented in the FHIR resource structure. In our use-case scenario, the FHIR resources are stored in the Mango database that we developed in our previous porotype. We developed various APIs on top of this database to retrieve the data stored in the FHIR resources format and perform data analytics. Figure 3 explains various sections of this framework and their connections. This framework has the following six major parts:FHIR databaseFHIR Query EngineMapping AlgorithmFHIR Compliant Database (Relational Database model)Analytic EngineUser Interface

### 4.1. FHIR Database

The FHIR database is the collection of FHIR resources that we already developed in our previous prototype and would be used as a dataset. Therefore, we are not discussing the creation of this database in this study. Within our database, we have different types of resources, each comprising a grand total of one hundred individual resources, but we utilized only those resources that are used for our data analysis.

### 4.2. FHIR Query Engine Layer

The FHIR query engine is a collection of FHIR queries, executing only FHIR queries based on core FHIR CRUD operations. Our query engine is responsible for accessing a list of available FHIR resources from the FHIR databases and preparing them for further processing. For this purpose, it uses the core FHIR RESTful APIs. Therefore, our query engine adeptly employs these RESTful APIs to extract and gather all FHIR resources in bulk out of the FHIR database and do some processing, filtering, and transformation within client-side code (in our case, the query engine). We leveraged the core FHIR GET and search APIs to access all resources from the FHIR database. The resulting data (FHIR resources) are assumed to be available in JSON format, the standard format for bulk FHIR data interchange. Table 2 shows the resulting data that have been retrieved from the FHIR database using REST APIs. For this purpose, we used an algorithm (see Algorithm 1) to access all FHIR resources housed within the database. Each type of resource has its own unique title and access parameters; therefore, for different FHIR resources, we used different resource names and search and access parameters within the resource URL to access each resource type. Figure 4 shows the block diagram of the query engine.
**Algorithm 1.** Algorithm to retrieved resources from FHIR database.1: **Function** Retrive_Resources()2: define resource type, e.g., patient3: define search parameters, e.g., resource id or any other attribute(s)4: value = Read resource id 5:    **while** (resources are available) **do**6:                    **GET** [base-url]/RsourceName?id = value7:     **end while**8: **end function** ** Retrive_Resources function **

#### Read (GET) Operation

The first feature provided by REST API is the read (GET) operation. This operation provided a way to access data and prepared it for further operations via various sub-operations. This standard FHIR API reads the FHIR resources from the databases or servers and transfers to the clients in the form of JSON.

The GET operation is designed to accept data extracted from the database or server via FHIR APIs operations. One of the primary functions of the GET request as a data request is a method to provide the data to the client. During the GET request operation, the clients (we) must provide the server or database with URLs indicating which data (data from resources) we seek to retrieve.

These URLs also enable us to receive updates on the operation’s progress and valuable information about retrieving the final results.

The retrieved data are made available to the client in a JSON format (in our case, the query engine). Figure 5 demonstrates that the query engine part reads the FHIR resources from the database using these APIs. Here is an example of the API query:

**GET** [base-url]/resource-type? parameters

For example, we could obtain data from a patient resource with identifier 23 using this query:

**GET** [base-url]/patient? identifier = 23

### 4.3. Mapping Agent/Algorithm

#### 4.3.1. Need of Mapping Algorithm

As aforementioned, the FHIR REST APIs are currently in their nascent stage, offering limited functionalities and operations that can be leveraged for healthcare data analytics applications. The FHIR REST APIs can only perform the core CRUD (Create, Read, Update, and Delete) operations, alongside a handful of other basic functionalities, on data stored in the FHIR resources. These operations are executed using standard mechanisms provided by the FHIR, and the REST APIs are happy to execute these operations on various FHIR resources while exchanging data between the FHIR server and the client.

However, the healthcare landscape is rapidly evolving, and there is an increasing demand for more advanced and complex operations on patient data in the healthcare environment, particularly in the FHIR data analytics domain. Additionally, healthcare analytics applications need to be improved to reduce the data processing burden and enhance the quality of data analyses [15]. Consequently, the REST APIs must evolve to prepare themselves for these challenges by incorporating more complex functionalities and executing more complicated queries. For this purpose, the FHIR offers standard mechanisms for extending API functionality, such as extension operations and search profiles.

However, certain types of operations, including data transformation, aggregations, search operations, and many more, are not currently achievable or impossible using the core FHIR APIs specification [15]. This limitation implies that executing more complex queries to perform advanced operations, such as any data analytics or analysis operations on clinical data stored in FHIR resources, is quite challenging and limited at this stage of REST. In other words, the core FHIR APIs encounter difficulties while performing data analytics directly on the patient data stored in the FHIR resources structure in the FHIR server or database. However, the use of data analytics in healthcare information systems is essential in the modern healthcare environment. As a result, we leveraged the FHIR core API functionalities and implemented a specialized intermediate layer known as a mapping algorithm/agent to simplify data analytics operations on the data stored in the FHIR resources.

#### 4.3.2. Role of Mapping Algorithm

The FHIR APIs present us with a wealth of resources, returned in the JSON format, which is a complex, hierarchical structure that nests data elements within tags. However, this structure is unsuitable for data analytics operations, which typically require structured or unstructured data, not data in a hierarchal structure [50]. Therefore, we must preprocess the JSON data by converting it into a tabular format and storing it in a FHIR-compliant relational database before applying any analytics.

For this purpose, we used a special agent that mapped the FHIR resource data into a format more suitable for data analytics. This mapping agent was responsible for converting the retrieved FHIR resource data via core FHIR APIs into a flat data format. The resulting data elements were then stored in a FHIR-compliant database, ready for analytics. The mapping algorithm is presented in Algorithm 2. Our mapping algorithm worked as a mapping agent between the FHIR API and FHIR-compliant database for data conversion. This mapping algorithm worked seamlessly for all types of resources in our dataset, for example, Patient, AllergyIntolerance, Practitioner, Condition, DiagnosticReport, ServiceRequest, Appointment, etc. Whenever we retrieved FHIR resource data from the FHIR centralized database, we applied the mapping algorithm on the way during FHIR API operations to retrieve and transform the data into the FHIR-compliant database; we named this data-mapping mechanism “Data Retrieval on Fly (DRF)”. The working mechanisms of this algorithm are illustrated in Figure 6, which depicts how it acted as a mediator between the FHIR API and a FHIR-compliant database, thus enabling the efficient conversion of hierarchical data into tabular data for analytics purposes.
**Algorithm 2.** Mapping Algorithm (Transform JSON data to EMR format).1: **Function** void main ()2:    Create Tables in MySQL database, once table for each resources type data and link these tables3:    Resource = Read (FHIR API resource)4:    Templet = Resource-Templet (Resource)5:    counter = Count(Temple)6:    **while** (counter > 0) **do**7:              **If** (Templet.Tag == Resource.Tag) **then**8:                            Table. attribute = Resource.Tag.Value9:              **end if**
10:
     counter = counter − 111: **end while**12: **end function** ** main function **13: ** This function used to compare Resource type **14: **Function** string Resource-Templet (Resource type)15: ** Create one dimension array for all resources and stored their tags. This is pre-defined templet for all resources **16: define string Result17: String Array List = [Patient, Condition, AllergyIntolerance, Practitioner, ServiceRequest, DiagnosticReport, Appointment, ………]18: String Patient [] = [“identifier”, “name”, “telecom”, “address“, “gender” …………]19: String Condition [] = [“identifier”, “clinical status”, “category”, “code” …………]20: String AllergyIntolerance [] = [“identifier”, “clinical status”, “code”, …………]21: String Practitioner [] = [“identifier”, “name”, “address”, “qualification”, …………]22: String DiagnosticReport [] = [“identifier”, “baseOn” status”, “category”, “code”,…….…]23: String ServiceRequest [] = [“identifier”, “baseOn” status”, “category”, “requester”,……]24: String Appointment [] = [“identifier”, “status”, “appointmentType”, “priority”, …………]25:    **If** (type == Patient) **then**26:                    Result = “Patient”27:        **else if** (type == Condition) **then**28:                                Result = “Condition”29:                 **else if** (type == AllergyIntolerance) **then**30:                               Result = “AllergyIntolerance”31:                   **else if** (type == Practitioner) **then**32:                                        Result = “Practitioner”33:                                **else if** (type == DiagnosticReport) **then**34:                                                       Result = ” DiagnosticReport”35:                                             **else if** (type == ServiceRequest) **then**36:                                                                     Result = “ServiceRequest”37:                                                          **else**38:                                                                  Result = “Appointment”39:      **end if**40: return (Result)41: **end function** ** Resource-Templet function **42: ** This function used to count the total number of tags in the resource **43: **Function** int Count(String Templet)44:    int counter = Templet.length45: return (counter)46: **end function** ** Count function **

### 4.4. FHIR Compliant Database

We created a special database called the FHIR-Compliant Database. This is a relational database schema with a collection of tables that have been designed to store the data represented in FHIR resources. The tables are connected with each other, and each table stores clinical data represented in the FHIR resources.

We have multiple resources, and each resource represents different types of clinical data. Therefore, first we created a table schema according to the data represented in the FHIR resources and logically connected these tables to facilitate the analytic query engine to query the data from multiple tables according to the workflows in the result generation. Each resource was stored in a single table or spread across multiple tables, for example, the patient resource data spread across multiple tables, etc. The table’s creation and connection were specifically designed to cater to the needs of the proposed workflows and required result generation.

Second, we applied a mapping algorithm that enabled us to retrieve the data elements from the FHIR resources and store them accurately in the corresponding tables in the FHIR-compliant database. The algorithm retrieved the data from the FHIR resources and then pushed it to the corresponding table. When querying the data from the FHIR database, the FHIR query engine utilizes RESTful APIs to read the resources in JSON format. On the way, the mapping algorithm seamlessly pre-processed this JSON data and transformed it to the relational database schema. This process is completed automatically, and all data from all FHIR resources are transformed into the sample EMR data format and stored in the relational tables. Figure 7 presents the FHIR-Compliant Database.

### 4.5. Data Analytic Engine

The data analytics layer plays a key role in this prototype. Once the FHIR resource data are seamlessly mapped to the relational database tables, they become ready for any data analytics operations. The data analytics is based on workflows (use-cases scenarios) that we have already designed for optimal results.

Our Data Analytics Engine (DAE) is a collection of selective SQL queries proficient in merging data from multiple tables, thereby providing unparalleled data analysis. We have created a series of distinct SQL queries, catering to our business use cases, which are then executed on the data stored in the SQL database to generate exceptional results. The queries have been designed in alignment with our workflows and expected outcomes.

The resulting data are unequivocally valuable and accessible to the end-users via an intuitive and efficient user interface. The detail-oriented results generated by the data analytics engine are undoubtedly the backbone of our prototype, providing insights into the data.

### 4.6. User Interface

The user interface of our framework is an elegant and sophisticated section, where the end-users access their desired data and obtain results catered specifically to their unique requirements. We developed a user-friendly graphical user interface to efficiently process data and generate results.

As a demonstration of the utility of our prototype, we developed an experimental data analysis user interface that shows the use of the search operations within the generic tool for exploring FHIR data sets. We created a number of FHIR data sets and a graphic visualization of these data sets that allowed for the demonstration of the data analytics on the clinical data used in healthcare settings and represented in the FHIR-based standard. The user interface of our prototype is presented in Figure 8.

## 5. Methods/Implementation

In this prototype or research work, we have stored our FHIR resource datasets in our NoSQL database (Mongo DB), which we had developed in our previous prototype. Therefore, we leverage the core FHIR APIs to perform data analytics on the data stored in these FHIR resources. We have employed the technique to download FHIR data into the FHIR-compliant database (SQL DB) and then applied data analytics to this FHIR data. For this purpose, we have utilized our developed mapping algorithm to transfer the FHIR resource data into the relational database tables. This has made it effortless to query data using standard SQL queries or tools and perform data analytics tasks on the data stored in these FHIR resources. It is essential to note that all the retrieval data from the FHIR resources require merging and formatting to support data analysis. As the patient’s unique clinical identifier is the key to connecting these objects, we have utilized this number to merge the data into a group of tables in a relational database to support further querying and analysis.

We have an extensive array of FHIR resources stored in our database; therefore, we have utilized the core FHIR GET and Search APIs to retrieve all the resources from the database. These APIs have seamlessly accessed the FHIR resources from the FHIR database, and we have performed various data analytics tasks depending on the defined use-cases. To provide our esteemed readers with a clear understanding of these APIs’ working mechanisms, we have discussed how FHIR APIs work for data analytics. Figure 2 illustrates a sample API operation mechanism in which the APIs access the clinical data and then perform the data analytics. This refers to the specialization of the FHIR API that focuses on providing the API’s functionality that is useful for healthcare data analytics applications.

This implementation has been executed in two phases:

**Phase 1:** We have developed various FHIR APIs to retrieve the FHIR resources from the FHIR database and then pre-process these resources using our developed algorithm to map the clinical data elements stored in the FHIR resource tags to a relational data model or schema and store the resulting data into the MySQL database. We have magnificently processed the FHIR resources via our algorithm, retrieved all data elements from these resources, and stored the result in database tables; we called it the FHIR-compliant database. For this purpose, we have crafted a database schema (tables) in the MySQL database (see Figure 7). Each resource type requires different parameters in the REST API URL to retrieve the FHIR resource from the database. Therefore, for each resource, we have provided a resource name and parameters depending on the resource type and data retrieval. For this purpose, we have executed an algorithm to perform this job for us. When the FHIR APIs retrieve resources from the FHIR database, on the way, the mapping agent/algorithm pre-processes the JSON format of FHIR resources and maps the data stored in various tags of JSON structure into the various MySQL database tables.

**Phase 2:** When the data were converted from a graph structure to a relational data model format, we applied various data analytics techniques to the data stored in the MySQL database. For this purpose, we have developed various types of SQL queries to generate our results. These SQL queries have impeccably matched the requirements of our use-cases, defined for our required data analytics. The output of these data analytics use-cases is shown in the Section 7. Figure 9 shows the implementation process. Furthermore, Figure 10 describes the complete framework process, including the techniques and computational tools applied in each step.

## 6. Experiments

We implemented our data analytics prototype/concept leveraging the FHIR database (Mongo DB), Python 3.9.8 programming language, and MySQL 5.6 database. We developed FHIR APIs, which enabled us to seamlessly retrieve various resources stored in the Mongo DB, consisting of a dataset size of 700 resources, inclusive of 100 resources of each resource type. Furthermore, before applying data analytics, we implemented our mapping algorithm/agent enabling the smooth transformation of FHIR resource tags to FHIR-compliant database (MYSQL) tables. We used the following:

Dataset Size: 700 resources (including 100 resources of each resource type)

Hardware: 4 Cores, 32 GB of RAM

Software: Windows 10 OS, Python 3.9.8 programming language, Mongo DB 4.4, 

MySQL 5.6 DB



Our experiment consisted of two phases:**Phase 1:** In this step, we implemented our FHIR APIs and executed algorithms to retrieve the FHIR resources from the Mongo DB. Furthermore, we also executed a mapping algorithm to transform the FHIR resource data into the relational database tables.**Phase 2:** In this step, we executed various SQL queries to perform highly precise data analytics based on the defined use-cases and generate the required results.

## 7. Results

To provide the underlying data for our esteemed results, we used the data stored in the relational data model, generated from the FHIR dataset stored in Table 2. The results are based on the use-cases we defined in our previous step. We have a number of use-cases, each based on a dataset different from others. Therefore, we executed various queries based on the use-cases. We generated various results from the FHIR dataset. We are discussing these use-cases and their results in detail here.

### 7.1. Use-Case 1

In this scenario, the queries used within the patient’s scalability count the number of patients that have been dutifully registered in the healthcare unit. These unparalleled queries seamlessly retrieve data from the patient table, which are associated with the esteemed patient resource in the FHIR dataset. Table 3 shows the retrieval data associated with patients gender-wise, and Figure 11 shows the graphical representation of this data.

### 7.2. Use-Case 2

In this scenario, the queries used within the patient’s scalability count the number of patients registered within the healthcare unit across a variety of years. These queries are specifically designed to retrieve relevant data from the patient table, which are closely associated with the patient resource within the FHIR dataset. The queries retrieved the data related to patients who have been registered within the healthcare system over a span of several years, ranging from the year 1950 to the year 2021. Table 4 presents the retrieval of the registered patients’ data in various years, while Figure 12 describes a graphical representation of this data.

### 7.3. Use-Case 3

In this particular scenario, the patient’s scalability has been measured by employing sophisticated queries aimed at counting the multitude of patients afflicted with diverse types of allergies. These queries were designed to extract relevant data from both the allergy and patient tables, which are associated with the Patient and AllergyIntolerance resources within the FHIR dataset. It joined data from these two tables because they belong to Patient and AllergyIntolerance resources and are spread across multiple tables and FHIR resources. Via these queries, relevant information relating to patients suffering from various allergies has been successfully retrieved. Table 5 presents the success of these queries, providing a comprehensive breakdown of the number of patients affected by different types of allergies. Furthermore, Figure 13 shows the graphical representation of this result.

### 7.4. Use-Case 4

In this particular scenario, the queries employed in the patient’s scalability quantify the number of distinct medical tests that have been requested by either a healthcare organization or a practitioner. These queries procure data from various tables, including patient, order, provider, Practitioner, etc., which are associated with the Patient, Practitioner, DiagnosticReport, and ServiceRequest resources in the FHIR dataset. The queries retrieved information related to various types of test orders that are present within the healthcare system. Table 6 represents the various types of medical tests undertaken by the patient. Additionally, the graphical representation of this data is illustrated in Figure 14.

### 7.5. Use-Case 5

In this scenario, the queries employed within the patient’s scalability counts the number of sundry categories of medical tests ordered by a healthcare organization or practitioner in different years. These queries procure data from various tables, including patient, order, provider, Practitioner, etc., which are associated with the Patient, Practitioner, DiagnosticReport, and ServiceRequest resources in the FHIR dataset. These queries have retrieved relevant information regarding assorted test orders in the healthcare system spanning a timeline from 1950 to 2021. The resulting outcome of these queries has been presented in Table 7, summarizing the diverse medical tests undertaken by the patient. Additionally, Figure 15 describes the graphical representation of this information.

## 8. Limitations

Our developed framework is capable of performing various types of descriptive data analytics on clinical data used in healthcare settings and represented in the FHIR-based standard. However, it is important to note that our study is limited in that it focuses solely on business use cases for patient clinical data belonging to two hospital information systems: patient registration systems and laboratory information systems. Other possible data analysis workflows and customized research scenarios based on patient data from other hospital information systems could be performed on FHIR-based data, but our current framework or tool does not directly support them without modification. In addition, there are some technical challenges in this research work:Our framework is currently developed under the FHIR R4 version and needs to be upgraded to the official FHIR R5 version when it gets finalized and released by HL7.Our framework might face issues in the coming FHIR version. HL7 FHIR specification requirements are changing over time, and the current resources might be replaced with any other new resources in the coming FHIR version. Additionally, the resource nature (from non-normative to normative) is changing over time. In this case, our framework might face challenges. Therefore, it needs to be updated in the coming FHIR versions if any of the mentioned cases happen. However, if none of these changes happen in the FHIR R5 version, it will work perfectly.Our framework executed multiple algorithms, such as the algorithm for accessing the FHIR resources via the RESTful APIs and the algorithm to map data from the FHIR resources to the EMR data format, and executed queries to perform data analytics for the end users. Therefore, the performance might not be ideal for every dataset. It worked excellently for our dataset (which is small), but the performance might be affected when dealing with large datasets, for example, when the number of resources and data elements in the dataset is in the billions or trillions.The interface of our framework works for our dataset (patient data used in patient registration systems and laboratory information systems); therefore, it would update if the workflow changed and included the data from other hospital information systems.

## 9. Discussion

In this study, we have developed an integrated framework or visual tool leveraging the cutting-edge FHIR standard, with prototype implementation and evaluation, aiming to empower standardized clinical statistics and analysis applications. This research work has three main contributions: First, the entire framework and workflow design follow the FHIR data standards, which could be reused for any other clinical data domain and could provide support for any clinical data that follows the FHIR standard. Second, the data analysis workflow and tools incorporate the experience of clinical researchers and statisticians and leverage powerful Python analytics, which could provide a starting point for FHIR researchers in this cutting-edge standard. Third, the intelligent mapping algorithm, artfully designed to facilitate the sublime process of data analytics or data analysis within the realm of FHIR-based data. The mapping algorithm could be reused for any other clinical data that follow the FHIR specification and need to process the FHIR-based data for other purposes, such as research or developing an artificial intelligence (AI) model or machine learning (ML) model, etc.

Our research effectively used the data-mapping algorithm for FHIR-based data to facilitate the data analytics process. Furthermore, mapped data could be utilized for other purposes, such as research, etc. Although recently, another technique, namely pathling [15], has been used for data analytics on FHIR-based data. However, it works in a limited domain because some operations are not easily or even currently possible to achieve via the FHIR REST API specification, such as data aggregation, searching the data, etc. Therefore, it is extremely challenging to implement. Furthermore, this technique is language-specific. Therefore, it needs to redesign the entire framework for a new language. Our technique is easy to implement and generally could be used for all FHIR-based data types and FHIR resources with minor modifications. Furthermore, the implementation process would work for every language.

Our developed framework or tool provides a graphical user-friendly interface to the end-users, such as healthcare professionals and researchers. The developed interface is used for FHIR data mapping and analytics purposes. Therefore, we developed two sub-menus, one for data mapping and a second for data analytic purposes (see Figure 8). However, we only discussed the data analytics sub-menu in this research work. The data mapping sub-menu is out of the scope of this study. Our data analytics sub-menu provided all options for our required results based on the defined use-cases. For example, the “Registered Patients” option provided results for all registered patients in the patient information systems. Similarly, “Test Order” generates the results of various types of patient laboratory tests ordered by any practitioner, healthcare organization, laboratory, etc. All the remaining options work accordingly. In short, it could greatly facilitate interactive, user-friendly data analysis.

In the future, we have a plan to extend our framework by adding data from other hospital information systems and updating the framework, including the data workflows and user interface, to make it more generic for users and researchers. Furthermore, we also intend to adopt the FHIR R5 version with particular COVID-19 and cancer-related resource definitions to represent COVID-19 and cancer data in our framework. This will help people working in the healthcare industry to enhance the consistency and quality of data analysis for cancer and COVID-19 data. Moreover, it will open more research dimensions for healthcare data analytic researchers in these areas.

## 10. Conclusions

In this study, we discussed the need for a data analytics tool to improve data analysis and reduce the skill burden in the healthcare industry. We have designed a comprehensive framework that empowers healthcare users (patients, practitioners, healthcare providers, etc.) to perform advanced data analysis on patient data used in healthcare settings and represented in the FHIR-based standard. The framework incorporates different data workflows based on patient data derived from two hospital information systems, namely patient registration systems and laboratory information systems, represented in the FHIR-based standard. Our use cases facilitate both patient-centered and cohort-based analysis and address common clinical user and researcher requirements. Although currently limited to two hospital information systems, the framework is flexible and can be extended to include data from other systems represented in the FHIR-based standard. With ongoing improvements, our framework will be valuable for healthcare applications in statistics and analytics. Overall, the goal of developing a state-of-the-art data analytics framework for clinical data in healthcare settings has been achieved.

## Figures and Tables

**Figure 1 healthcare-11-01729-f001:**
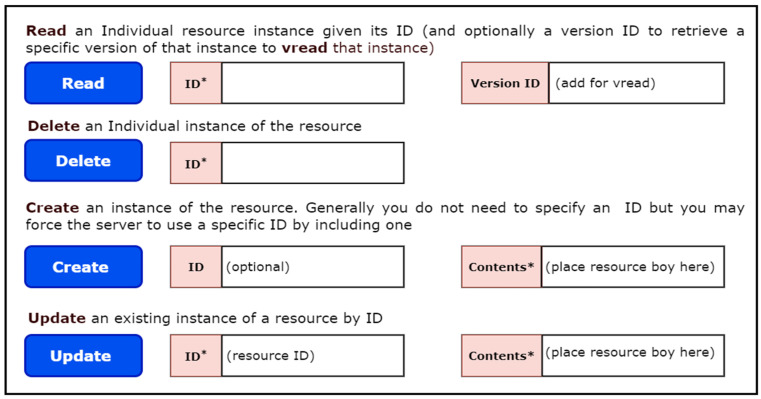
CRUD operations of the patient resource in the FHIR server.

**Figure 2 healthcare-11-01729-f002:**
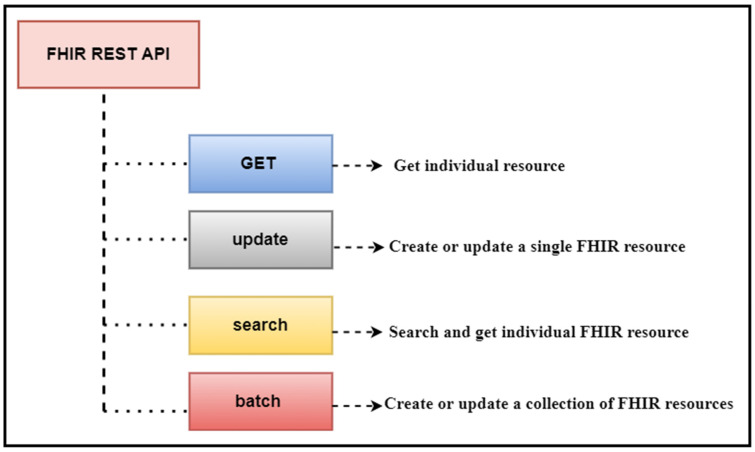
REST APIs Operations: The operations performed on clinical data represented in FHIR resources.

**Figure 3 healthcare-11-01729-f003:**
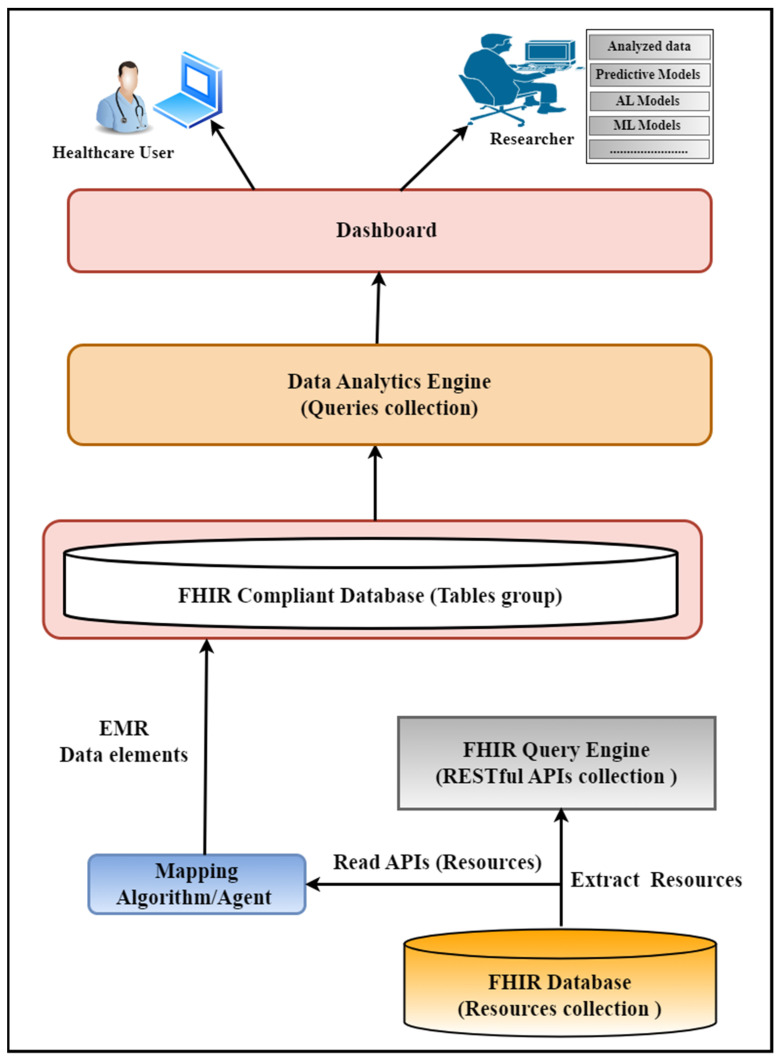
Block diagram of the proposed data analytic framework: Explains various sections of the framework and their connections.

**Figure 4 healthcare-11-01729-f004:**
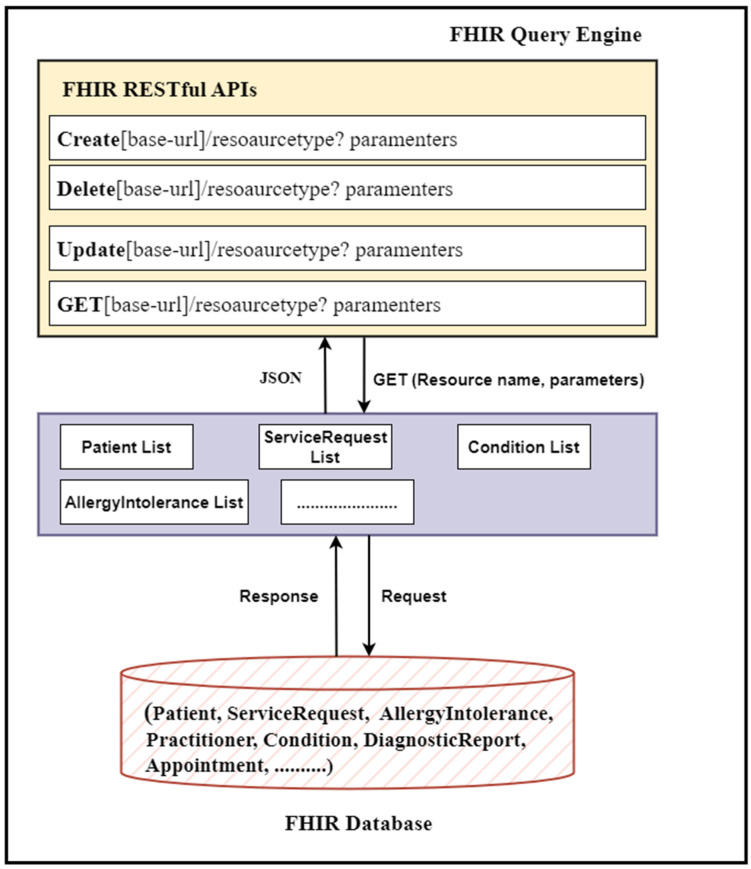
Query engine working mechanism: Query engine read FHIR resources in bulk.

**Figure 5 healthcare-11-01729-f005:**
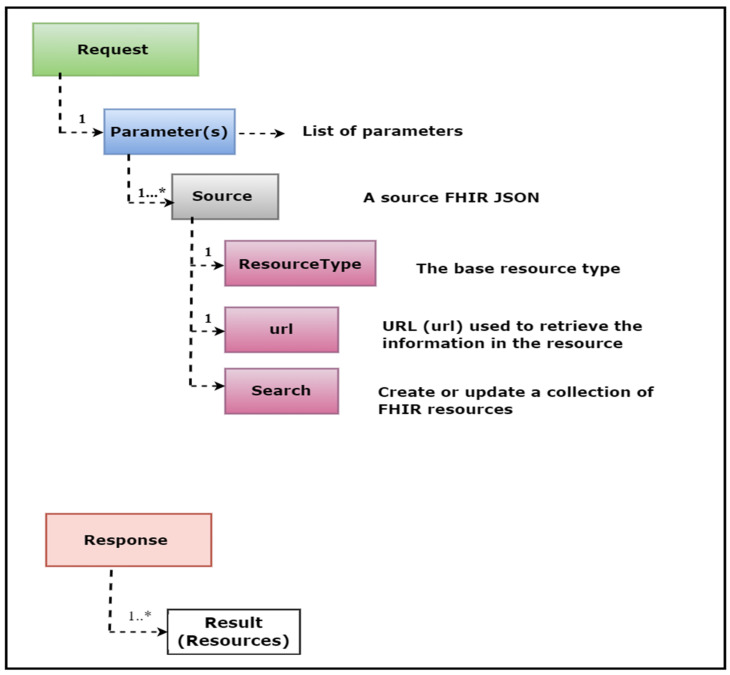
The GET APIs to extract resources from FHIR database.

**Figure 6 healthcare-11-01729-f006:**
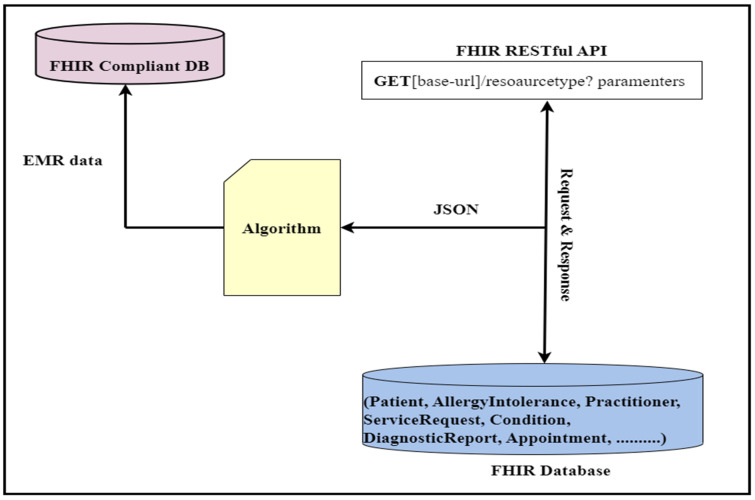
Mapping algorithm working mechanism.

**Figure 7 healthcare-11-01729-f007:**
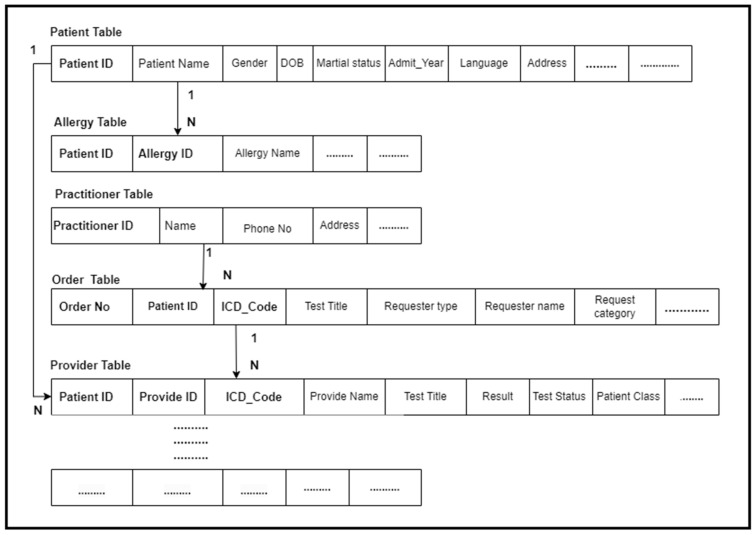
FHIR-compliant database: A sample schema of compliant database.

**Figure 8 healthcare-11-01729-f008:**
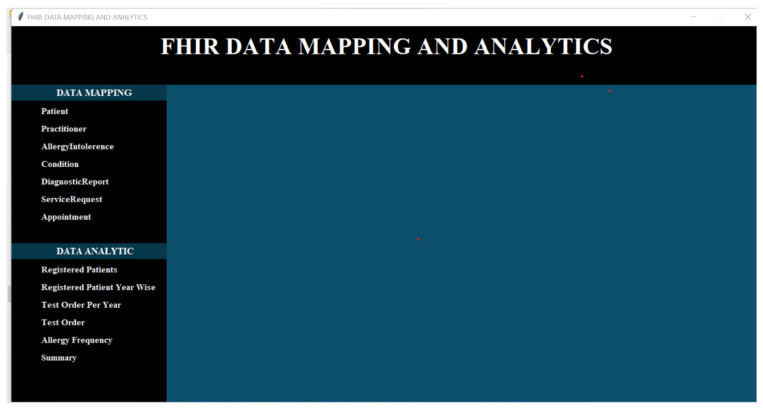
Experimental user interface for data analytics.

**Figure 9 healthcare-11-01729-f009:**
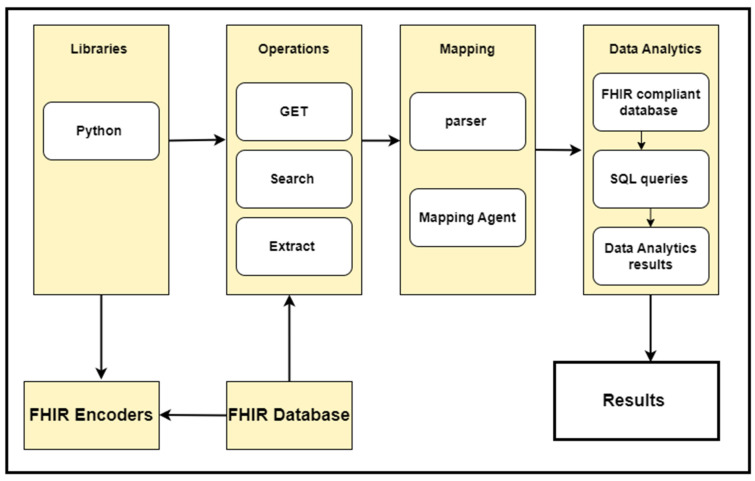
Described the implementation components and process.

**Figure 10 healthcare-11-01729-f010:**
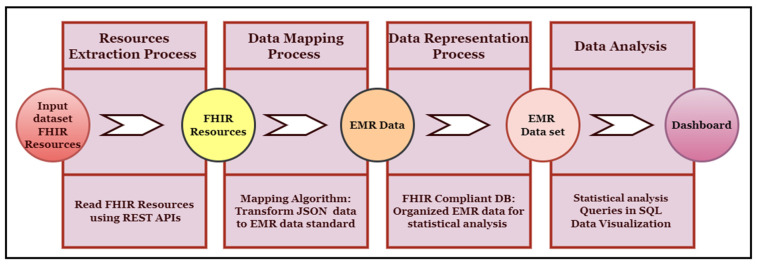
Framework working process: Describes each step working and implementation processing.

**Figure 11 healthcare-11-01729-f011:**
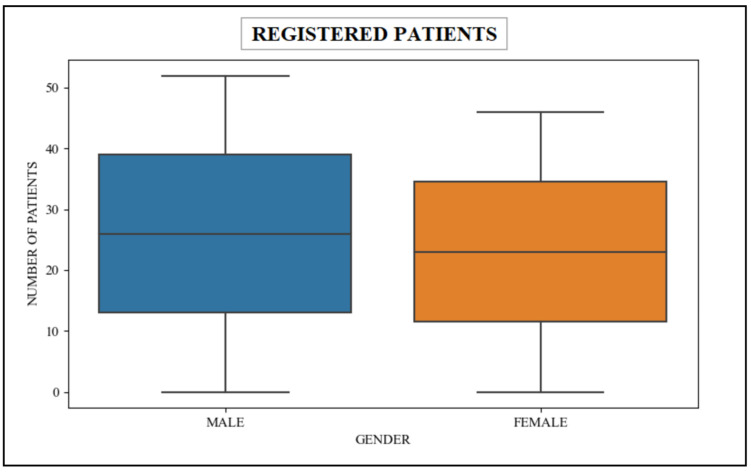
Registered patients Gender-wise (patient-centered-based analysis).

**Figure 12 healthcare-11-01729-f012:**
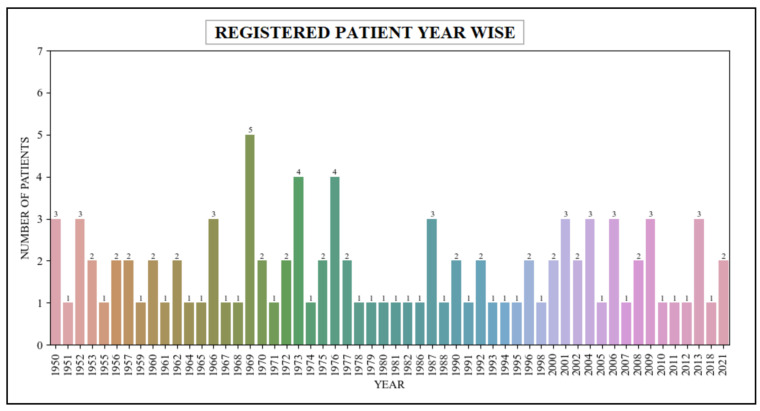
Registered patients within a specified timeframe (patient-centered-based analysis).

**Figure 13 healthcare-11-01729-f013:**
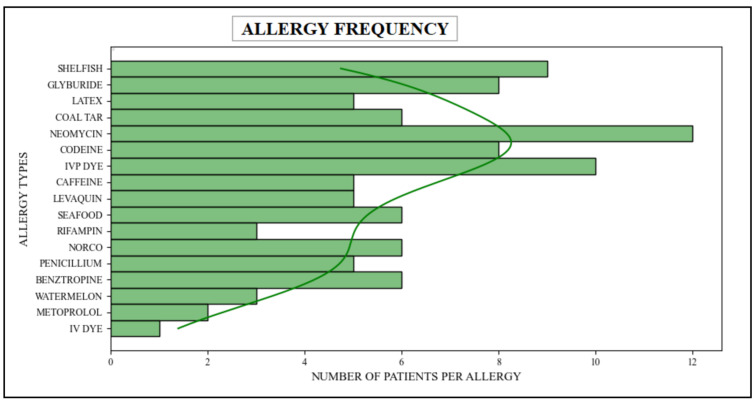
Patients and various types of allergies association (cohort-based interactive analyses).

**Figure 14 healthcare-11-01729-f014:**
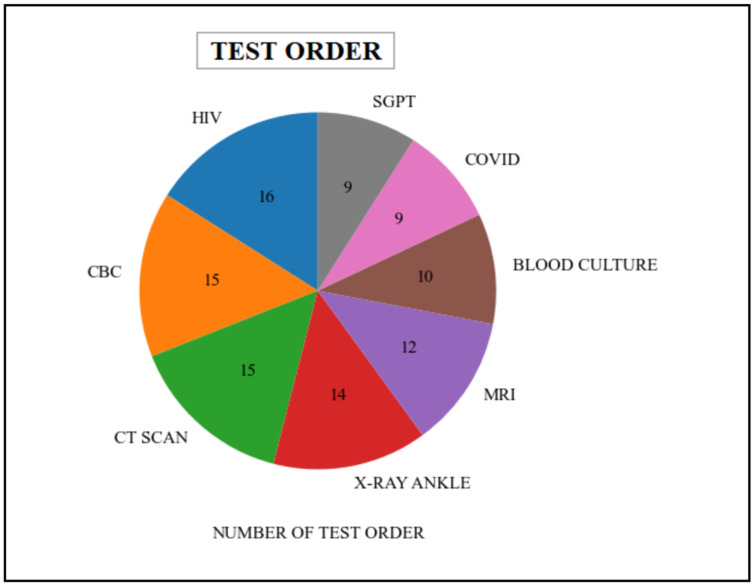
Patient various types of medical test orders (cohort-based interactive analyses).

**Figure 15 healthcare-11-01729-f015:**
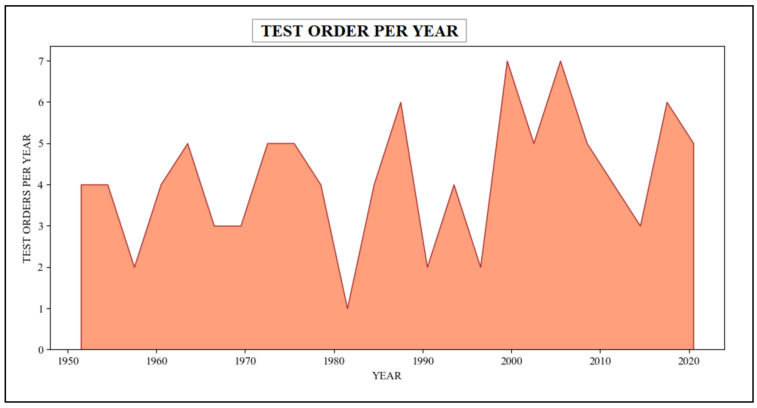
Patient various types of medical test orders within a specified timeframe (cohort-based interactive analyses).

**Table 1 healthcare-11-01729-t001:** List of data analysis workflows (business use-cases) for patient data in the healthcare setting.

No	Descriptions
1	Investigate registered patients in healthcare settings
2	Investigate registered patients in healthcare settings within a specified timeframe
3	Investigate patients having various types of allergies
4	Investigate various types of tests ordered by a physician, organization, etc.
5	Investigate various types of tests ordered by a physician, organization, etc., within a specified timeframe

**Table 2 healthcare-11-01729-t002:** FHIR resources retrieved from FHIR database using APIs.

No	Resource Type	Total Resources
1	Patient	100
2	AllergyIntolerance	100
3	Practitioner	100
4	ServiceRequest	100
5	DiagnosticReport	100
6	Condition	100
7	Appointment	100

**Table 3 healthcare-11-01729-t003:** Registered patients gender-wise (patient-centered-based analysis).

Male	Female
55	45

**Table 4 healthcare-11-01729-t004:** Registered patients within a specified timeframe (patient-centered-based analysis).

Year’s	1950	1951	1952	1953	1955	------	2013	2018	2021
**Patient’s No**	3	1	3	2	1	------	3	1	2

**Table 5 healthcare-11-01729-t005:** Number of patients having various types of allergies (cohort-based interactive analyses).

No	Allergy	No’s of Patients
1	Shellfish	9
2	Glyburide	8
3	Latex	5
4	Coal Tar	6
5	Neomycin	12
6	Codeine	8
7	IVP Dye	10
8	Caffeine	5
9	Levaquin	5
10	Seafood	6
11	Rifampin	3
12	Norco	6
13	Penicillium	5
14	Benztropine	6
15	Watermelon	3
16	Metoprolol	2
17	IV Dye	1

**Table 6 healthcare-11-01729-t006:** Patient various types of medical test orders (cohort-based interactive analyses).

Test Name	HIV	CBC	CT SCAN	X-Ray Ankle	MRI	Blood Culture	COVID	SGPT
**Tests order Percentage**	16	15	15	14	12	10	9	9

**Table 7 healthcare-11-01729-t007:** Patient various types of medical test orders within a specified timeframe (cohort-based interactive analyses).

Year’s	1951	1952	1953	1955	----	2010	2015	2019	2020
**Number of tests order**	4	4	3	2		5	3	6	5

## Data Availability

The datasets used or analyzed in this study are available from the corresponding author on reasonable request.

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
