# Peer review of "Transforming Healthcare Analytics with FHIR: A Framework for Standardizing and Analyzing Clinical Data"

_healthcare, 2023, doi:10.3390/healthcare11121729_

Round 1
Reviewer 1 Report
Thank you for submitting your paper to the Healthcare: MDPI journal. I have read your entire paper and saw many issues in it that need to be addressed before resubmitting your paper.
1. Your abstract section is really inappropriate, and it's your fault if your research doesn't reflect in the abstract section. In addition, in the abbreviated section, they abruptly discussed
FHIR Model. Please mention the full form of this model the first time. It's very important.
2. The flow of the introduction is not okay. The motivation section is presented well, but they miss adding proper information and expressing their motivation in this section. The contribution needs to be added to the introduction section; that's very important to mention.
3. The proposed model does not define how they are analyzing the data; if it's based on a model, show the model's performance and efficacy. Authors need to show mathematically
4. Please compare your work with some recent publications. It's a very common study, and I hope the author will clarify the novelty of their work.
5. Reference formatting is not okay; need too fix it.
Author Response
"Please see the attachment"

Reviewer 2 Report
1can you explain the FHIR Data Analytics Framework in your research article
2. Literature Review: Add recent years' review, as many lot improvements were made in the recent years
3 Conclusion: reduce the size of the conclusion
4. reduce the size of the abstract, it should be precise
Author Response
"Please see the attachment"

Reviewer 3 Report
Thank you to the authors for preparing this interesting paper. Data analytics is an important growing area important to industry, particularly health analytics. The authors present a framework for performing analytics on clinical data in healthcare.
This is a well written paper the whose findings are based on empirical evidence and make an original contribution.
The introduction provides relevant background and establishes a need for this work, and defines key terms.
The literature review points out gaps in current health care data analysis. After reading this the reader has an idea of the key achievements in health care analytics. Nice work.
The materials section outlines the process followed.
There are many diagrams provided in the findings, some of these are great and are accompanied by a clear and useful explanation such as Figure 3. However some digrams do not appear to add to the discussion, for example Figure 8, it is hard to read and does not really give the reader a feel for the interface designed. A more detailed explanation of how the framework was tested and the theoretical basis would be beneficial.
In the discussion the authors outline the shortcomings of the work and include the implications for practice and theory.
Author Response
"Please see the attachment"

Reviewer 4 Report
The article presents a framework for standardizing and analyzing clinical data in the healthcare industry using FHIR, a scalable standards-based data model. While the article highlights the potential benefits of using FHIR and the proposed framework, there are also limitations and challenges to be considered.
Pros:
- The framework leverages the FHIR standard, which could be reused in other clinical data domains and enable extensive support for any clinical data that follows the FHIR specification.
- The workflow design follows the FHIR-based standard, which promotes consistency and interoperability.
- The framework incorporates the experience of clinical statisticians and researchers, leveraging powerful Python analytics, which could greatly facilitate interactive, user-friendly data analysis.
- The framework can perform analytics on clinical data in healthcare settings and by healthcare data analytics researchers.
- The experimental results demonstrate the proposed framework's ability to generate various analytics from clinical data represented in the FHIR resources.
Cons:
- The study is limited in that it focuses solely on business use cases for patient clinical data belonging to two hospital information systems, patient registration systems and laboratory information system. Other possible data analysis workflows and customized research scenarios on the patient data belong from other hospital information systems could be performed upon on FHIR-based data but the current framework or tool does not directly support them without modification.
- The framework is currently developed under FHIR R4 version and needs to be upgraded to the official FHIR R5 version when it gets finalized and released by HL7.
- The framework might face issues in the coming FHIR version because the HL7 FHIR specification requirements are changing overtime and the current resources might be replaced with any other new resources in the coming FHIR version.
- The performance of the framework might be not ideal for every dataset, especially when dealing with large datasets, for example, when the number of resources and data elements in the dataset are in millions or trillions.
- The interface of the framework is working for the dataset used in patient registration system and laboratory information systems, but it would need to be updated if the workflow changes and includes data from other hospital information system.
Overall, the proposed framework is a promising tool for standardizing and analyzing clinical data in the healthcare industry. However, it is important to note the limitations and challenges mentioned in the article. The authors acknowledge the need for further development and enhancements of the framework to address these limitations and to make it more useful for healthcare users and researchers. Future work includes extending the framework to incorporate data from other hospital information systems and adopting the FHIR R5 version with particular COVID-19 and cancer-related resources definitions to represent COVID-19 and cancer data in the framework.
Need for improvement:
1. Table 1 format
2. Figure 1 contrast
3. Table 2 format
4. Figure 5 - Font type are little incomprehensible
5. Table 6, thru 9 format
6. Figure 16 is illegible even though its a summary it should be legible enough
Author Response
"Please see the attachment"

Round 2
Reviewer 1 Report
The authors failed to respond to all comments properly, and they must clarify the proposed model.
Writing style should improve, and authors need to conduct literature reviews properly.
Model comparisons need to be clarified with other published papers. The novelty of the work is really low.
Compare with: https://ieeexplore.ieee.org/abstract/document/10063857
https://link.springer.com/article/10.1007/s11334-022-00523-w
https://www.sciencedirect.com/science/article/pii/S2772662223000851
Comments 3:
|
Answer: Thank you for your valuable comments. We already explained that our model is based on our defined Use cases. We discussed in details that it is descriptive data analytics, which is based on user define use-cases NOT on any model. It does not based on any model like machine learning or predictive model, etc. Therefore, we did not mentioned that how our model analyized the data and also did not test the model performance. We believe that it would be clear for the readers. |
But it's a model, need to analysis properly the proposed model.
Author Response
"Please see the attachment."
